# *Bartonella* spp. Prevalence (Serology, Culture, and PCR) in Sanitary Workers in La Rioja Spain

**DOI:** 10.3390/pathogens9030189

**Published:** 2020-03-04

**Authors:** Aránzazu Portillo, Ricardo Maggi, José A. Oteo, Julie Bradley, Lara García-Álvarez, Montserrat San-Martín, Xavier Roura, Edward Breitschwerdt

**Affiliations:** 1Departamento de Enfermedades Infecciosas, Hospital U. San Pedro-Centro de Investigación Biomédica de La Rioja (CIBIR), 26006 Logroño, La Rioja, Spain; aportillo@riojasalud.es (A.P.); jaoteo@riojasalud.es (J.A.O.); lgalvarez.ext@riojasalud.es (L.G.-Á.); 2Galaxy Diagnostics, Research Triangle Park, Morrisville, NC 27709, USA; rgmaggi@ncsu.edu; 3Intracellular Pathogens Research Laboratory, Comparative Medicine Institute, College of Veterinary Medicine, North Carolina State University, Raleigh, NC 27695, USA; Julie_Bradley@ncsu.edu; 4Faculty of Social Sciences, University of Granada, 52006 Melilla, Spain; momartin@ugr.es; 5Hospital Clínic Veterinari, Universitat Autònoma de Barcelona, 08193 Bellaterra, Barcelona, Spain; xavier.roura@uab.cat

**Keywords:** *Bartonella* Alpha-Proteobacteria Growth Medium (BAPGM), *B. henselae*, *B. quintana*, *B. vinsonii* subsp. *berkhoffii*, *B. koehlerae*, sanitary workers

## Abstract

*Bartonella* spp. are increasingly implicated in association with a spectrum of zoonotic infectious diseases. One hundred sanitary workers in La Rioja, Spain, completed a questionnaire and provided blood specimens for *Bartonella* spp. serology and *Bartonella* Alpha-Proteobacteria growth medium (BAPGM) enrichment blood culture/PCR. Six immunofluorescence assays (IFA) were performed and aseptically obtained blood specimens were inoculated into liquid BAPGM and subcultured onto blood agar plates. *Bartonella* DNA was amplified using conventional and real-time PCR assays. The *Bartonella* spp., strain, or genotype was determined by DNA sequencing. *Bartonella* seroreactivity was documented in 83.1% and bloodstream infection in 21.6% of participants. *Bartonella henselae*, *B. vinsonii* subsp. *berkhoffii* genotypes I and III, and *B. quintana* were identified. IFA seroreactivity and PCR positivity were not statistically associated with self-reported symptoms. Our results suggest that exposure to and non-clinical infection with *Bartonella* spp. may occur more often than previously suspected in the La Rioja region.

## 1. Introduction

The genus *Bartonella* comprises fastidious Gram-negative, slow growing and facultative intracellular bacteria belonging to the Alpha-2 subgroup of the class Proteobacteria, Order Rhizobiales. These microorganisms are most often transmitted to humans through animal bites or scratches (cats, dogs and other animals), or by scratch inoculation of infected flea and body louse feces [1]. In addition, sand-fly vector-competence was experimentally proven for transmission of *Bartonella bacilliformis* in Peru during the last century [2]. Other arthropods such as ticks, head lice, bedbugs, bat flies and mites have been associated with human *Bartonella* spp. infections, but their role as competent vectors requires further confirmation [3,4,5,6,7]. *Bartonella* spp. can survive within different hosts and reservoirs (arthropods, mammals, humans) for months to years. This genus has been increasingly associated with a wide spectrum of zoonotic emerging and reemerging infectious diseases. The number of validated species (more than 35), potential species (at least 17) and subspecies (at least three) continues to increase <http://www.bacterio.net/bartonella.html>. Some species, such as *B. bacilliformis*, cause potentially life threatening illness, but are limited geographically by the transmitting vector, whereas other species such as *B. quintana* are transmitted under poor hygienic conditions throughout the world. Flea-transmitted *B. henselae* is the most frequent etiological agent of sub-acute and chronic lymphadenopathy named cat-scratch disease (CSD) in children and teenagers and occurs throughout the world. A subset of CSD patients develop severe or systemic disease manifestations, including endocarditis, osteomyelitis, granulomatous hepatitis and hepatosplenic abscess [8,9,10,11,12]. Apart from these ‘classical’ species, at least 15 *Bartonella* spp. have been associated with human diseases (e.g., *Bartonella vinsonii* subsp. *berkhoffii* or *B. vinsonii* subsp. *arupensis*) or, at least, have been detected in humans (e.g. *B. vinsonii* subsp. *vinsonii*).

Seemingly, all *Bartonella* spp. may represent potential opportunistic pathogens for infections in animals and human patients [13,14]. Under a One Health prism, the collective understanding of *Bartonella* epidemiology and pathogenesis continues to change [13]. After recent studies in Brazil documented *Bartonella* infection (positive PCR/DNA sequencing results) in healthy blood donors [15,16,17], the authors initiated this study to assess the prevalence of *Bartonella* spp. infection in sanitary workers in a localized region of Spain. Previously, a high prevalence of *Bartonella* spp. antibodies, as well as DNA of these bacteria, were reported in blood samples collected from veterinary workers in Spain [18]. Compared to veterinary workers, who are frequently exposed to arthropod vectors and infected animals, sanitary workers in La Rioja were not considered to be at a high risk for exposure to or infection with *Bartonella* spp. In addition to microbiological testing, the potential relationship of *Bartonella* spp. antibodies or blood stream infection (as assessed by DNA amplification and sequencing) with minor or nonspecific, self-reported symptoms, such as fatigue or insomnia was examined.

## 2. Results

### 2.1. Subject Recruitment

Study participants were recruited from among physicians, nurses, researchers, medical students, and administrative personnel who worked at the Center of Biomedical Research (CIBIR) or San Pedro’s University Hospital at La Rioja (Northern Spain). The buildings housing these facilities are adjacent to each other and personnel share a common café. Physicians, medical students and nurses have contact with patients diagnosed with CSD or other *Bartonella* infections. The CIBIR houses the Center of Rickettsiosis and Arthropod-borne Diseases, where researchers support the diagnosis of arthropod (especially tick borne, but also flea, mite, and mosquitoes) associated diseases. Clinical specimens are tested using serological and molecular assays for *Bartonella*, *Borrelia*, *Coxiella*, *Rickettsia* spp. and other pathogens. *Bartonella* spp. are not routinely cultured and are infrequently isolated. Isolates of *B. quintana* and *B. vinsonii* subsp. *berkhoffii* genotypes have not been cultured in the research laboratories.

After excluding three individuals who had recently taken antimicrobials, 97 sanitary workers were included in the study. A total of 194 participant samples (97 EDTA-blood and 97 sera) and the accompanying questionnaires were used to generate data reported in the results. Questionnaire demographic and risk exposure characteristics are provided in Table 1. Trekking was the only parameter statistically associated with PCR negativity.

### 2.2. Bartonella Seroprevalence and Blood Stream Infection

#### Serology

Seroreactivity among the six *Bartonella* spp. or genotypes ranged from 16.5% to 62.9% with the lowest percentage reactivity to *B. quintana* (Table 2). Only 16 of 97 (16.5%) participants were not seroreactive to any test antigen. Seroreactivity to *B. henselae, B. vinsonii* subsp. *berkhoffii* genotypes I, II and III, and *B. koehlerae* ranged from 20.6% to 62.9%. Among individual study participants, seroreactivity patterns varied among the six *Bartonella* spp. or genotypes used for immunofluorescence assay (IFA) testing (Table 2). All but three *Bartonella* spp. bacteremic individuals were seroreactive to at least one test antigen. *Bartonella* spp. IFA seroreactivity was not statistically associated with any specific symptom.

### 2.3. BAPGM Enrichment Blood Culture PCR

Twenty-one participants (21.6%) had a positive *Bartonella* alpha-Proteobacteria growth medium (BAPGM) enrichment blood culture/PCR result (Table 3). *Bartonella henselae*, *B. vinsonii* subsp. *berkhoffii* genotype I and *B. vinsonii* subsp. *berkhoffii* genotype III DNA was amplified and sequenced directly from extracted blood DNA in four, three, and two individuals, respectively (Table 4). An additional individual was also positive for *Bartonella* DNA amplification from blood, but species identification was not possible. No direct evidence (PCR/DNA sequencing) of *Bartonella vinsonii* subsp. *berkhoffii* genotype II, *B. quintana* or *B. koehlerae* was found among participant blood DNA extractions. Following BAPGM enrichment blood culture, sixteen participants were positive for *Bartonella* DNA amplification or bacterial isolation. Four of nine participants, who were PCR positive following blood DNA extraction (three *B. vinsonii* subsp. *berkhoffii* genotype I, and a single *B. henselae*), were again PCR/DNA sequence positive for the same organism following BAPGM enrichment blood culture. The species identified in blood cultures were *B. henselae* (six participants), *B. vinsonii* subsp. *berkhoffii* genotype I (six participants), *B. vinsonii* subsp. *berkhoffii* genotype III (four participants), and *B. quintana* (one participant) (Table 4). Co-infections with *B. quintana* and *B. vinsonii* subsp. *berkhoffii* genotype III, or *B. henselae* and *B. vinsonii* subsp. *berkhoffii* genotype III were each detected in a single participant. *Bartonella quintana* and *B. henselae* blood agar plate isolates were obtained from two and one participant, respectively (Table 3). PCR sequencing results for the 16S-23S ITS region, *groEl, rpoB* and 16S rRNA genes for each isolate are provided in Table 4.

### 2.4. Assay Associations (Serology and BAPGM Enrichment Blood Culture-PCR)

Seventeen of 21 bacteremic individuals were seroreactive to at least one *Bartonella* sp. antigen (three seronegative participants were PCR positive in blood culture and/or by bacterial isolation). Interestingly, two *B. henselae* bacteremic patients were not *B. henselae* seroreactive, but one each was seroreactive to *B. vinsonii* subsp. *berkhoffii* genotypes II and III, or *B. vinsonii* subsp. *berkhoffii* genotype II and *B. koehlerae* (Table 5). BAPGM enrichment blood culture/PCR positivity was not associated with self-reported symptoms. Corticosteroid intake, demographics, and arthropod or animal exposures were not statistically associated with BAPGM enrichment blood culture/PCR positivity (Table 1). Information about exposures, demographics, serologic assays and clinical features of the individuals with BAPGM enrichment blood culture/PCR positivity are summarized in Table 6.

## 3. Discussion

In this study, conducted among ‘healthy volunteers’ working in a sanitary setting in Spain, 83.1% participants were *Bartonella* seroreactive and 21.6% had molecular evidence of bloodstream infection using the BAPGM enrichment blood culture-PCR platform. These high percentages are unprecedented in the few published studies testing healthy participants against a large (6 *Bartonella* strains) IFA serology panel, as well as the high prevalence of occult bloodstream infection among individuals from the Rioja region of Spain [15,16]. Using the same IFA assays, a previous study of 32 healthy medical personnel from North Carolina identified only one *B. henselae* seroreactor (antibody titer 64); however, 16 (50%) individuals were *B. vinsonii* subsp. *berkhoffii* genotype II seroreactive [19]. Among these North Carolina healthy individuals, no participant was *B. vinsonii* subsp. *berkhoffii* genotype I or III, *B. koehlerae,* or *B. quintana* seroreactive and *Bartonella* spp. DNA was not PCR amplified from any blood or BAPGM enrichment blood culture using the same diagnostic platform used in this study [19,20]. Testing of stored blood samples from these 32 individuals over a decade later by droplet digital PCR (ddPCR) also resulted in negative *Bartonella* spp. DNA results (Maggi R. unpublished data). Technically, the same personnel (Bradley and Maggi) performed the serology and BAPGM enrichment blood culture testing in the current and previous North Carolina study. Based upon the results of this study, *Bartonella* spp. occupational exposure risk in sanitary workers should be further investigated.

Interestingly, *B. vinsonii* subsp. *berkhoffii* genotype II seroreactivity was unexpectedly high among healthy sanitary workers in both studies (62.9% Spain, and 50% North Carolina). In the only other study involving healthy individuals and the BAPGM enrichment blood culture platform, Brazilian medical personnel screened 500 blood donors [15,16]. Seroprevalence was 16% and 32%, respectively for *B. henselae* and *B. quintana* (IFA antigen slides provided by the Centers for Disease Control and Prevention), and blood stream infection was found in 16 (3.2%) of the 500 donors, infected with either *B. henselae* (n = 15) or *B. clarridgeiae* (n = 1). It is important to point out that similar to the current study, IFA serology was not consistently associated with bloodstream infection. Recent documentation of bloodstream infection among healthy individuals has further increased the diagnostic complexity associated with the genus *Bartonella*. As fastidious Gram-negative bacteria, *Bartonella* spp. require several weeks to grow axenically. Despite efforts to improve yield, obtaining agar plate isolates remains difficult and successful isolation has only been reported in a small number of human cases, often requiring microbiological techniques limited to specialty laboratories. Isolation is more likely to occur when culturing immunocompromised individuals, who presumably maintain higher levels of bacteremia [21]. Serological assays (IFA) are universally used for diagnosis of *Bartonella* infection in animals and humans, although sensitivity has been considered poor [22,23]. In the context of IFA specificity, cross reactions with other bacterial pathogens such as *Coxiella burnetii*, *Chlamydia* spp., *Rickettsia* spp. (spotted fever group), *Treponema pallidum, Orientia tsutsugamushi, Francisella tularensis, Ehrlichia chaffeensis, Mycoplasma pneumoniae,* and *Escherichia coli* have been reported in human patients [24,25]. In context of One Health, *Bartonella* spp. IFA has a high degree of specificity (97% or greater depending upon IFA antigen) when testing dogs [26,27,28]. Sera from dogs with very high IFA titers (8192) following experimental infection with *Rickettsia rickettsii* do not induce fluorescence (cross reactivity) to *Bartonella* spp. antigens [29]. However, similar to humans, *Bartonella* IFA assays have poor sensitivity when testing sera from PCR/culture positive dogs [30]. The extent to which sequential or simultaneous exposure to multiple *Bartonella* spp. or the frequency of *Bartonella* intergenus IFA specificity contributes to serology results in dogs and humans, when using a broad panel of antigens that differ from the classical *B. henselae* and *B. quintana* diagnostic assays, awaits additional clarification. Dogs also developed *Bartonella* species-specific IFA antibodies following experimental infection with *B. henselae* or *B. vinsonii* subsp. *berkhoffii* [26,27]. Similarly, a veterinarian developed *B. vinsonii* subsp. *berkhoffii* genotype-specific antibodies after an inadvertent needle stick transmission [31]. Overall, evidence to support *Bartonella* spp. IFA cross-reactivity in dogs to closely related alpha-Proteobacteria or other more distantly related bacteria is lacking and the extent to which co-exposures, rather than cross-reactivity, contributes to serology findings in humans remains unclear. For people who live in places like La Rioja where free time is often spent in rural environments, co-exposures may be more frequent than previously suspected.

Demonstration of *Bartonella* spp. DNA in blood or tissues using PCR assays can be technically limiting due to low level infection (minimal target DNA), a relapsing bacteremia, or *Bartonella* PCR primer designs that differ from the numerous *Bartonella* spp. genotype and strain targets (failure to anneal and amplify). Thus, negative PCR results do not exclude the presence of authentic *Bartonella* infections, as previously described [21]. Histological examination (Warthin-Starry staining) of a biopsy specimen at a site of systemic involvement is often not sufficiently specific to establish a diagnosis of *Bartonella* infection since other infectious diseases (e.g., tularemia) cannot be readily distinguished. Although immunohistochemical analyses are less sensitive than PCR amplification techniques, their usefulness to establish the etiologic diagnosis of *Bartonella* infections has been reported, but these assays are available in only a few laboratories [32].

Recently, specialized culture techniques based on growth enrichment in modified media combined with PCR assays and subculture bacterial isolation (BAPGM platform) have been developed with the aim of enhancing documentation of *Bartonella* infection [33,34]. Validation of the BAPGM enrichment blood culture/PCR platform for the assessment of *Bartonella* spp. bloodstream infection in dogs was originally reported in the *Journal of Microbiological Methods* [35,36]. Subsequently the BAPGM platform has been used diagnostically to assess bloodstream infection in dogs, other animal species, [37,38] and in humans [19,20,39,40,41]. It is clear that the developmental and microbiological utilization of more sensitive or robust microbiological techniques such as MALDI TOF, Next Generation Sequencing and specialized enrichment/PCR techniques enhance the detection of microorganisms in various patient samples [42,43,44].

At times, these more sensitive techniques detect organisms in healthy individuals that were previously associated with pathology in sick individuals, which is seemingly historically applicable to the genus *Bartonella*. Increased detection sensitivity can be a benefit to the patient and physician, but as illustrated by the results of this study, increased sensitivity can also complicate clinical decision-making, as well as microbiological and pathological interpretations. In this study, statistically significant associations were not detected between serology or the detection of *Bartonella* bloodstream infection and self-reported symptoms, such as fatigue and insomnia.

In the current study, BAPGM enrichment blood culture/PCR resulted in the amplification of *B. vinsonii* subsp. *berkhoffii* genotype III DNA from four participants. In contrast to the results of this study, human studies from the United States using the BAPGM platform most often result in amplification of *B. vinsonii* subsp. *berkhoffii* genotype II and less frequently genotype I DNA [19]. To our knowledge, human infection with *B. vinsonii* genotype III has not been reported from North America, whereas this genotype has been identified in dogs and humans in several Mediterranean countries, including Spain [18,45,46]. Interestingly, the genotype III was also documented in military working dogs with endocarditis imported to the United States from Europe [47]. In addition to all PCR negative controls remaining negative throughout the study, failure to detect *B. vinsonii* subsp. *berkhoffii* genotype II, in conjunction with the detection of genotype III provides further support for a lack of laboratory contamination with PCR products or organisms. In addition, the *B. henselae* and *B. quintana* sequences obtained in this study were most consistent with sequences reported by European, as compared to North American investigators (Table 4 and analysis by Maggi RG).

The results of this study have created additional diagnostic challenges for clinicians attempting to confirm a diagnosis of bartonellosis in an individual patient. For example, microbiological criteria for a clinical diagnosis of *Bartonella* endocarditis in a patient with a blood culture-negative endocarditis (BCNE), include *Bartonella* spp. PCR positive or an IFA antibody titer of ≥800 against *B. henselae* or *B. quintana* antigens [48]. Amplification of *B. henselae* or *B. quintana* DNA from excised heart valves also establishes a diagnosis of *Bartonella* endocarditis in BCNE cases [9,48]. The prevalence of *Bartonella* infection using serological techniques is also a cause of concern. In the present study, *Bartonella* seroprevalence in healthy participants (>83%) was substantially higher than the prevalence’s reported from blood donors from the same area nearly 20 years before (<6%), and even exceeded previous seroprevalence values from risk groups, such as cat owners (28.9%) and HIV-infected people (17.3%) [49,50]. Clearly, differences in the number of *Bartonella* spp. antigens and variable sensitivity among individual IFA antigens used at these two times are likely contributors, but it is also possible that the epidemiology of *Bartonella* spp. transmission has changed in the region. In support of this possibility, using the same antigens, the seroprevalence of sanitary workers was more than twice the seroprevalence found among veterinary sanitary workers from different regions of Spain according to a study performed during the same year [18]. The results of this study only marginally impact the clinical interpretation of results for BCNE patients; however, our findings complicate interpretation of serology and PCR results for patients with non-specific symptoms, chronic lymphadenitis, granulomatous hepatitis and other forms of pathology, particularly when *Bartonella* DNA is amplified, but antibody reactivity is low or not detected. Obviously, a single serological result (an IgG seroreactive value) does not confirm an acute or chronic infection, since IgG antibodies may be due to a prior exposure to the microorganism.

Recent documentation of the historical co-evolutionary efficiency of *Bartonella* spp. among mammalian reservoir hosts located throughout much of the world represents an amazing and evolving area of research [51,52]. In addition to transmission by a substantial number of documented and suspected vectors, *Bartonella* spp. have been transmitted by needle stick, animal bites and scratches and potentially by blood transfusion [31,53]. Thus, our epidemiological understanding of this genus continues to evolve; thereby influencing medical understanding of transmission patterns among animals and human patients. When bartonellosis is clinically suspected, a diagnostic approach that incorporates results of serology, enrichment culture and molecular techniques should always be interpreted in the context of the medical history, exposure risk and within the differential diagnosis that excludes other microorganisms.

## 4. Materials and Methods

### 4.1. Study Design and Subject Recruitment

A cross-sectional study was performed to determine the seroprevalence to six *Bartonella* species/genotypes. Bacteremia was concurrently assessed by means of *Bartonella* alpha-Proteobacteria growth medium (BAPGM) enrichment blood culture platform. Institutional review board approval for this study was obtained from the Ethical Committee of Clinical Research from La Rioja (CEICLAR) in January 29, 2016 (Ref. CEICLAR PI-209). 

### 4.2. Data and Specimen Collection

A standardized questionnaire including demographic information, symptoms experienced, domestic and wild animal bites, scratches or exposures, and travel history, was completed. Exposure to, or bites by different arthropods (lice, fleas, ticks, mites, bed bugs and others) was recorded. Approximately 10–12 mL of blood (5–6 mL in EDTA, 5–6 mL in a serum separator tube) was collected at the time of enrollment. Aseptic blood collection, including chlorhexidine decontamination of the skin, was performed by an experienced nurse. Three participants, who reported antimicrobial use within the last 2 months on the questionnaire, were subsequently excluded from the study.

### 4.3. Specimen Processing and Diagnostic Testing

Refrigerated EDTA-anticoagulated blood and serum samples were processed in less than two hours at the Center of Rickettsiosis and Arthropod-Borne Diseases (CRETAV), located at the Center for Biomedical Research from La Rioja (CIBIR, Logroño, La Rioja, Spain), where blood was centrifuged and sera stored at −80 °C until prepared for shipping to Galaxy Diagnostics, Inc., Research Triangle Park, North Carolina, USA.

### 4.4. Bartonella IFA Serological Testing

*Bartonella vinsonii* subsp. *berkhoffii, B. henselae, B. koehlerae* and *B. quintana* antibodies were determined in the Intracellular Pathogens Research Laboratory (IPRL) at North Carolina State University (North Carolina, USA) using cell culture grown bacteria as antigens and following standard immunofluorescent antibody assay (IFA) techniques [18,19]. Canine isolates of *B. vinsonii* subsp. *berkhoffii* genotype I (NCSU 93CO-01 Tumbleweed, ATCC type strain #51672), *B. vinsonii* subsp. *berkhoffii* genotype II (NCSU 95CO-08, Winnie) and *B. vinsonii* subsp. *berkhoffii* genotype III (NCSU 06CO-01 Klara), feline isolates of *B. henselae* SA2 strain (NCSU 95FO-099, Missy) and *B. koehlerae* (NCSU 09FO-01, Trillium) and *B. quintana* (NCSU11-MO-01 Monkey origin) were passed from agar plate grown cultures into *Bartonella*-permissive cell lines, i.e., the DH82 (a canine monocytoid) cell line for strains *B. henselae* SA2, *B. quintana, B. vinsonii* subsp. *berkhoffii* I and *B. koehlerae* and Vero cells (a mammalian fibroblast cell line) for *B. vinsonii* subsp. *berkhoffii* II and III to obtain antigens for IFA testing. For each antigen, heavily infected cell cultures were spotted onto 30-well Teflon-coated slides (Cell-Line/Thermo Scientific), air-dried, acetone-fixed, and stored frozen. Fluorescein conjugated goat anti-human IgG (Cappel, ICN) was used to detect bacteria within cells using a fluorescent microscope (Carl Zeiss Microscopy, LLC, Thornwood, NY). Serum samples diluted in a phosphate-buffered saline (PBS) solution containing normal goat serum, Tween-20, and powdered nonfat dry milk to block nonspecific antigen binding sites were first screened at dilutions of 1:16 to 1:64. All sera that were reactive at a reciprocal titer of 64 were further tested with two-fold dilutions out to 1:8192. To avoid confusion with possible nonspecific binding found at low dilutions, a cutoff of 64 was selected as a seroreactive titer.

### 4.5. Growth Medium

Enrichment blood culture was performed at Galaxy Diagnostics Inc., Research Triangle Park, North Carolina, USA, as previously described [18,19,20]. An aliquot of 1 mL of EDTA whole blood was inoculated into 10 mL of BAPGM, after which the cultures were maintained at 35 °C in a 5% CO_2_, water-saturated atmosphere. After 7, 14, and 21-day incubation periods, PCR was performed on each inoculated liquid culture and a 1 mL aliquot of the enrichment culture was inoculated onto blood agar plates and incubated at 35 °C. Plates were checked for colony formation at 7, 14, and 21 days after plating. An un-inoculated BAPGM culture (negative control) was processed in an identical manner with each group of study participant specimens.

### 4.6. Conventional and Real-Time PCR Analysis

DNA was extracted using standard operating procedures from EDTA anticoagulated blood, enrichment liquid blood cultures incubated for 7, 14, and 21 days, and from blood agar plate colony isolates, if obtained after subculture from BAPGM enriched blood specimens [19]. *Bartonella* spp. DNA was amplified using primers designed to amplify two distinct consensus sequences in the *Bartonella* 16S-23S intergenic spacer (ITS) region as described previously with minor modifications [18]. Two sets of oligonucleotides, 325s and 1100as were used as forward and reverse primers for conventional PCR, and primers 325s and 543as were used as forward and reverse primers for quantitative PCR along with TaqMan probe 438 (Table 1). Additionally, as previously reported [18], conventional PCR screening for *B. koehlerae* was performed using species-specific oligonucleotides Bkoehl-1s and Bkoehl-1125as as forward and reverse primers, respectively (Table 1). Amplification of the ITS region at both genus and species (*B. koehlerae*) levels were performed in a 25 μL final volume reaction containing 12.5 μL of either MyTaq HS Red Mix 2X (Bioline) for *B. koehlerae* conventional PCR or Sso Advanced Universal Probe Supermix (BioRad) for *Bartonella* genus real-time PCR; 0.2 μL of 100 μM of each forward and reverse primer (IDT-DNA Technology), 7.3 μL of molecular-grade water, and 5 μL of DNA from each sample tested. PCR negative controls were prepared using 5 μL of dH_2_O (when testing isolates from plates), 5 μL of DNA from blood of a healthy dog, or 5 μL of DNA extracted from un-inoculated BAPGM-negative controls (when testing BAPGM enrichment cultures). Positive controls for PCR were prepared by serial dilution (using dog blood DNA) of genomic DNA from *B. henselae* (Houston I strain type) down to 0.001 pg/μL (equivalent to 0.5 bacteria/μL). Conventional PCR was performed in an Eppendorf Mastercycler EP gradient under the following conditions—a single hot-start cycle at 95 °C for 3 min followed by 55 cycles of denaturing at 94 °C for 15 seconds (s), annealing at 66 °C for 15 s, and extension at 72 °C for 18 s. Amplification was completed by an additional cycle at 72 °C for 1 min, and products were analyzed by 2% agarose gel electrophoresis with detection using ethidium bromide under ultraviolet light. Amplicon products were sequenced to determine the *Bartonella* species and ITS strain type. Real-time PCR was performed in an CFX96 Real-time System (Bio-Rad) under the following conditions: a single hot-start cycle at 95 °C for 3 min followed by 44 cycles of denaturing at 94 °C for 10 s, annealing at 66 °C for 10 s, and extension at 72 °C for 10 s. Fluorescence at channel 1 was detected during the extension cycle. As in conventional PCR, all amplicon products were sequenced to determine the *Bartonella* species and ITS strain type. All PCR and uninoculated BAPGM enrichment controls remained negative throughout the study period. PCR assays targeting *GroEl*, *rpoB* and 16S rRNA genes were also performed to confirm identification of agar plate isolates (Table 7).

### 4.7. Data Analysis

Questionnaire data for the study were collected on paper forms and entered into a Microsoft Excel database. Data entry was validated by comparison of the electronic records with the information on the forms. Associations of demographic, risk factor, symptoms and exposure variables were assessed with means and medians for continuous variables, and with counts and rates in contingency tables for categorical data. Group comparisons were performed. The Fisher exact test for categorical variables and the Mann–Whitney U test for non-categorical variables were used. Data processing was carried out with the R software [18], version 3.3.1 for Windows.

## Figures and Tables

**Table 1 pathogens-09-00189-t001:** Demographic and exposure histories for 97 sanitary workers from Spain, comparing statistical differences between *Bartonella* PCR-positive and PCR-negative individuals.

Demographics and Travel	PCR+ n = 19 (%)	PCR− n = 78 (%)	U/OR	95% CI	*p*-Values
Age (years) Median Mean Minimum Maximum	35.0 36.8 21 55	39.5 41.2 21 64	U = 527.5		0.174
**Gender** Female Male	13 (68.4%) 6 (31.6%)	53 (67.9%) 25 (32.1%)	0.978	0.272–3.169	1
**Housing** Urban Peri-urban Rural area/farm Rural area/forest	15 (78.9%) 4 (21.1%) 0 (0%) 0 (0%)	62 (79.5%) 11 (14.1%) 3 (3.8%) 1 (1.3%)			0.798
**Clinical condition** Healthy Persistent/chronic disease Infectious disease	17 (89.5%) 1 (5.3%) 2 (10.5%)	66 (84.6%) 13 (16.7%) 8 (10.3%)	1.539 0.280 1.029	0.297–15.463 0.006–2.118 0.098–5.852	0.731 0.291 1
**Clinical features** Persistent fever Fatigue Insomnia Sleepiness Memory problems Headache Irritability Anxiety Depression Tremor Vision impairment Eye pain Balance problems Bladder dysfunction Shortness of breath Tachycardia Poor appetite Weight gain Chronic diarrhea Corticosteroid treatment	0 (0%) 2 (10.5%) 0 (0%) 1 (5.3%) 0 (0%) 3 (15.8%) 0 (0%) 0 (0%) 0 (0%) 0 (0%) 0 (0%) 0(0%) 0 (0%) 0 (0%) 0 (0%) 1 (5.3%) 0 (0%) 1 (5.3%) 0 (0%) 2 (10.5%)	3 (3.8%) 4 (5.1%) 8 (10.3%) 1 (1.3%) 1 (1.3%) 16 (20.5%) 3 (3.8%) 2 (2.6%) 0 (0%) 2 (2.6%) 0 (0%) 2 (2.6%) 1 (1.3%) 1 (1.3%) 1 (1.3%) 2 (2.6%) 2 (2.6%) 3 (3.8%) 1 (1.3%) 5 (6.4%)	0 2.156 0 4.191 0 0.729 0 0 - 0 - 0 0 0 0 2.092 0 1.384 0 1.707	0–10.152 0.181–16.520 0–2.388 0.052–339.02 0–159.708 0.121–3.029 0–10.152 0–22.196 - 0–22.196 - 0–22.196 0–159.708 0–159.708 0–159.708 0.034–42.27 0–22.196 0.025–18.428 0–159.708 0.150–11.573	1 0.334 0.349 0.355 1 0.758 1 1 - 1 - 1 1 1 1 0.484 1 1 1 0.620
**Allergy** Autoimmune disease Dried fruits Metals Food Animals Lactose Mites Pollen	3 (15.8%) 0 (0%) 0 (0%) 0 (0%) 0 (0%) 2 (10.5%) 0 (0%) 1 (5.3%) 1 (5.3%)	26 (33.3%) 1 (1.3%) 2 (2.6%) 3 (3.8%) 0 (0%) 1 (1.3%) 0 (0%) 4 (5.1%) 12 (15.4%)	0.378 0 0 0 - 8.762 - 1.027 0.308	0.065–1.498 0–159.708 0–22.196 0–10.152 - 0.433–538.98 - 0.020–11.243 0.007–2.358	0.169 1 1 1 - 0.097 - 1 0.453
**Pets** Dogs Cats Birds	13 (68.4%) 10 (52.6%) 6 (31.6%) 3 (15.8%)	50 (64.1%) 35 (44.9%) 26 (33.3%) 21 (26.9%)	1.211 1.361 0.924 0.512	0.377–4.333 0.442–4.252 0.257–2.983 0.087–2.063	0.794 0.613 1 0.387
**Arthropod exposure** Fleas Ticks Lice Bed bugs	4 (21.1%) 4 (21.1%) 3 (15.8%) 0 (0%)	32 (41.0%) 29 (37.2%) 20 (25.6%) 5 (6.4%)	0.387 0.454 0.547 0	0.085–1.367 0.100–1.612 0.093–2.213 0–4.559	0.121 0.280 0.549 0.580
**Animal exposure** Dogs Cats	4 (21.1%) 3 (15.8%)	17 (21.8%) 10 (12.8%)	0.957 1.272	0.204–3.571 0.202–2.756	1 0.715
**Animal scratches and/or bites** Dogs Cats Birds	2 (10.5%) 4 (21.1%) 1 (5.3%)	15 (19.2%) 17 (22.8%) 5 (6.4%)	0.497 0.957 0.813	0.050–2.488 0.204–3.571 0.016–7.945	0.511 1 1
**Outdoors activities** Trekking Hunting Fishing Agriculture Gardening	5 (26.3%) 0 0 (0%) 0 (0%) 2 (10.5%)	44 (56.4%) 0 3 (3.8%) 8 (10.3%) 19 (24.4%)	0.280 - 0 0 0.368	0.072–0.924 - 0–10.152 0–2.388 0.038–1.792	**0.022** - 1 0.349 0.231
**Travel out of Spain** Other European countries North America Central America South America Asia Africa Australia/New Zealand	15 (78.9%) 8 (42.1%) 7 (36.8%) 2 (10.5%) 4 (21.1%) 4 (21.1%) 0 (0%)	66 (84.6%) 28 (35.9%) 28 (35.9%) 16 (20.5%) 13 (16.7%) 8 (10.3%) 5 (6.4%)	0.685 1.295 1.041 0.459 1.329 2.309 0	0.173–3.319 0.401–4.030 0.310–3.270 0.047–2.277 0.277–5.167 0.450–10.081 0–4.559	0.510 0.609 1 0.512 0.737 0.243 0.580

Percentages may not total 100% if participants checked more than one category. U is statistic of the Mann–Whitney test. OR is the odds ratio. Percentages are relative to the group (PCR+ or PCR−).

**Table 2 pathogens-09-00189-t002:** Immunofluorescent antibody (IFA) titers to six *Bartonella* spp. or genotypes for 97 sanitary workers tested for *Bartonella* exposure. Numerical values represent the number of titers at various dilutions and the total number of seroreactors to each antigen.

	*Bartonella* Antigen					
IFA Titer	*Bh* SA2	*Bq*	*Bvb* TI	*Bvb* TII	*Bvb* TIII	*Bk*
**<64**	45	81	77	36	42	51
**64**	18	11	6	26	27	26
**128**	18	4	10	23	15	13
**256**	12	1	4	11	12	5
**512 or 1024**	4	0	0	1	1	2
*** Seroreactive** **%**	52	16	20	61	55	46
	53.60%	16.50%	20.60%	62.90%	56.70%	47.40%

*Bh* SA2: *Bartonella henselae* San Antonio 2 strain; *Bq*: *Bartonella quintana*; *Bvb* TI: *Bartonella vinsonii* subsp. *berkhoffii* genotype I; *Bvb* TII: *Bartonella vinsonii* subsp. *berkhoffii* genotype II; *Bvb* TIII: *Bartonella vinsonii* subsp. *berkhoffii* genotype III; *Bk*: *Bartonella koehlerae*. * Total seroreactive: number and % of individuals with titers ≥ 64.

**Table 3 pathogens-09-00189-t003:** Blood and *Bartonella* alpha-Proteobacteria growth medium (BAPGM) enrichment blood culture PCR/DNA sequencing results for 97 sanitary workers from Spain.

Participants (n = 97)	Species and Strain				
Sample Type	*Bh* SA2	*Bvb* TI	*Bvb* TIII	*Bq*	Any Species
**Blood**	4	3	2	0	9
**7-day culture**	0	2	4	0	6
**14-day culture**	3	1	0	0	4
**21-day culture**	3	3	0	1	7
**Agar plate isolates**	1	0	0	2	3
*** Total positive participants (%)**	9 (9.3%)	6 (6.2%)	6 (5.2%)	2 (2.1%)	21 (21.6%)

*Bh*: *Bartonella henselae* San Antonio 2 strain; *Bvb* TI: *Bartonella vinsonii* subsp. *berkhoffii* genotype I; *Bvb* TIII: *Bartonella vinsonii* subsp. *berkhoffii* genotype III; *Bq*: *Bartonella quintana*. * Total % bacteremic.

**Table 4 pathogens-09-00189-t004:** Sequence identity comparisons for the three *Bartonella* agar plate isolates obtained following subculture.

Isolate	Bacterial Species	ITS Region		*GroEl* Gene		*rpoB* Gene		16S rRNA	
		GenBank	bp (%)	GenBank	bp (%)	GenBank	bp (%)	GenBank	bp (%)
		ID		ID		ID		ID	
GL-90 *Bq*	*Bq* Toulouse	BX897700	541/541 100	BX897700	525/525 100	BX897700	593/593 100	BX897700	870/870 100
GL-92 *Bq*	*Bq* Toulouse	BX897700	559/559 100	BX897700	525/525 100	BX897700	593/593 100	BX897700	877/877 100
GL-96 *Bh*	*Bh* Houston I	CP020742	507/536 94.7	CP020742	519/519 100	AF171070	600/600 100	CP020742	878/878 100
	*Bh* SA2	AF369529	536/536 100	AF304021	482/482 * 100	NA *		NA *	

bp: base pairs; *Bq*: *Bartonella quintana*; *Bh*: *Bartonella henselae*. * NA partial or no comparable sequence data in GenBank for *B. henselae* San Antonio 2 (SA2) strain.

**Table 5 pathogens-09-00189-t005:** IFA serology results for sanitary workers that were *Bartonella* PCR positive.

Sanitary Worker ID/*Bartonella* spp. or Genotype Sequenced		IFA titers					
ID	*Bartonella* DNA PCR	*Bh*	*Bq*	*Bvb* TI	*Bvb* TII	*Bvb* TIII	*Bk*
	Species/Genotype						
GL-6	***Bvb* TIII**	**256**	**128**	**128**	**256**	**256**	**128**
GL-17	***Bh***	<16	<16	32	**64**	**64**	<16
GL-18	***Bh***	**256**	**64**	32	**64**	**128**	**64**
GL-21	***Bh***	**512**	<16	<16	32	32	16
GL-26	***Bh***	**128**	16	<16	16	16	<16
GL-32	***Bvb* TIII *+ Bh***	**512**	<16	<16	<16	**64**	**128**
GL-37	***Bvb* TI**	**64**	**256**	**128**	**256**	**64**	**128**
GL-47	***Bvb* TIII**	<16	<16	<16	32	32	32
GL-49	***Bvb* TI**	**64**	16	16	**64**	**256**	**64**
GL-58	***Bvb* TI**	32	<16	32	**256**	**256**	**1024**
GL-65	***Bh***	16	<16	<16	**64**	32	**64**
GL-80	***Bvb* TI**	**256**	**128**	**256**	**512**	**256**	**256**
GL-84	***Bh***	**256**	**128**	**256**	**256**	**256**	**256**
GL-86	***Bvb* TI**	<16	<16	32	**64**	32	32
GL-87	***Bh***	**128**	**64**	**128**	**128**	**64**	**64**
GL-90	***Bq***	32	<16	<16	<16	16	<16
GL-92	***Bvb* TIII *+ Bq***	<16	<16	<16	<16	16	32
GL-96	***Bh***	**256**	**64**	**256**	**256**	**128**	**128**
GL-98	***Bvb* TIII**	<16	<16	16	32	**64**	<16
GL-100	***Bvb* TI**	**64**	<16	**128**	**256**	**64**	**64**

ID: identification number; *Bh*: *Bartonella henselae*; *Bq*: *Bartonella quintana*; *Bvb* TI: *Bartonella vinsonii* subsp. *berkhoffii* genotype I; *Bvb* TII: *Bartonella vinsonii* subsp. *berkhoffii* genotype II; *Bvb* TIII: *Bartonella vinsonii* subsp. *berkhoffii* genotype III; *Bk*: *Bartonella koehlerae*.

**Table 6 pathogens-09-00189-t006:** Demographics, health status and exposures of sanitary workers with BAPGM enrichment blood culture/PCR positivity.

Bartonella spp. by BAPGM + PCR	Age	Sex	Health Status	*Bartonella* IFA Sero-Reactivity	Living Area	Cat Exposure	Dog Exposure	Other Animal Exposure	Outdoor Activities	Arthropod Exposure
***Bvb* TIII** 6	34	F	Healthy	**Pos** *Bvb, Bq, Bh, Bk*	Urban	Yes (bite)	Yes (bite)	Rodent	Gardening	No
***Bh*** 17	40	M	Healthy	**Pos** *Bvb*	Urban	No	Yes	No	Trekking Cycling Diving	Fleas-Ticks-Biting flies-Mosquitoes-Spiders
***Bh*** 18	43	F	Healthy	**Pos** *Bvb, Bq, Bh, Bk*	Urban	Yes	No	Rodent	No	No
***Bh*** 21	39	M	Healthy	**Pos** *Bh*	Urban	Yes	No	Bird Reptile Hedgehog	No	Fleas-Ticks-Mosquitoes-Spiders
***Bh*** 26	31	F	Healthy	**Pos** *Bh*	Urban	Yes	No	No	Trekking	Fleas-Ticks-Spiders
***Bvb* TIII + *Bh*** 32	33	F	Healthy	**Pos** *Bvb, Bh, Bk*	Peri-urban	Yes	Yes	Poultry	No	Mosquitoes
***Bvb* TI** 37	21	F	Healthy	**Pos** *Bvb, Bq, Bh, Bk*	Urban	No	Yes	No	No	No
***Bvb* TIII** 47	33	M	Healthy	Neg	Urban	No	No	No	Cycling	No
***Bvb* TI** 49	21	F	Healthy	**Pos** *Bvb, Bh, Bk*	Urban	No	No	No	No	No
***Bvb* TI** 58	27	F	Healthy	**Pos** *Bvb, Bk*	Urban	Yes (bite)	Yes (bite)	No	Trekking	Fleas- Biting flies-Lice-Mites
***Bh*** 65	N.A.	F	N.A.	**Pos** *Bvb, Bk*	Peri-urban	No	No	No	No	No
***Bvb* TI** 80	31	F	Healthy	**Pos** *Bvb, Bq, Bh, Bk*	Urban	No	No	No	Cycling	Mosquitoes
***Bh*** 84	48	F	Healthy	**Pos** *Bvb, Bq, Bh, Bk*	Urban	No	No	No	No	No
***Bvb* TI** 86	36	M	Healthy	**Pos** *Bvb*	Urban	No	Yes	No	No	No
***Bh*** 87	52	F	Healthy	**Pos** *Bvb, Bq, Bh, Bk*	Urban	No	No	No	No	No
***Bq*** 90	30	M	Healthy	Neg	Urban	Yes (bite)	Yes	Bird Poultry Pigs Sheep	Trekking Cycling Diving	Fleas-Ticks-Biting flies-Mosquitoes-Lice-Spiders
***Bvb* TIII + *Bq*** 92	38	M	Healthy	Neg	Urban	No	Yes	No	No	Biting flies-Mosquitoes-Spiders
***Bh*** 96	55	F	Healthy	**Pos** *Bvb, Bq, Bh, Bk*	Peri-urban	No	Yes	No	No	No
***Bvb* TIII** 98	43	F	Healthy	**Pos** *Bvb*	Urban	Yes (bite)	No	No	Trekking	Ticks- Biting flies-Mosquitoes-Lice-Spiders
***Bvb* TI** 100	50	F	Healthy	**Pos** *Bvb, Bh, Bk*	Peri-urban	No	Yes (bite)	Bird Rabbit Rodent	Trekking Gardening Cycling Diving	Ticks- Biting flies-Mosquitoes-Lice

F: female; M: male; *Bq*: *Bartonella quintana*; *Bvb TIII*: *Bartonella vinsonii* subsp. *berkhoffii* genotype III; *Bh*: *Bartonella henselae*; *Bvb TI*: *Bartonella vinsonii* subsp. *berkhoffii* genotype I; N.A.: Data not available.

**Table 7 pathogens-09-00189-t007:** Primers and probe used for PCR testing in this study.

Oligonucleotide	Type	Sequence
**BsppITS325s**	Sense primer	5’ CCTCAGATGATGATCCCAAGCCTTCTGGCG 3’
**BsppITS543as**	Antisense primer	5’ AATTGGTGGGCCTGGGAGGACTTG 3’
**BsppITS1100as**	Antisense primer	5’ GAACCGACGACCCCCTGCTTGCAAAGCA 3’
**BsppITS438**	TaqMan probe	5’ FAM-AGGTTTTCC/ZEN/GGTTTATCCCGGAGGGC-IABkFQ 3’
**Bkoehl-1s**	Sense primer	5’ CTTCTAAAATATCGCTTCTAAAAATTGGCATGC 3’
**Bkoehl1125as**	Antisense primer	5’ GCCTTTTTTGGTGACAAGCACTTTTCTTAAG 3’
**Sequencing analysis**		

PCR amplicon DNA sequencing was performed by a commercial company (Genewiz, Research. Triangle Park, NC). Chromatogram evaluation and sequence alignments were performed using ContigExpress and AlignX software (Vector NTI Suite 10.1, Invitrogen Corp., Carlsbad, CA). Bacteria species and genotype were defined by comparing similarities with other sequences deposited in the GenBank database using the Basic Local Alignment Search Tool (Blast v. 2.0), and an in-house curated database (Align X, Vector-NTI-Invitrogen).

## Data Availability

Data supporting the conclusions of this article are included in the article. To assure participant confidentiality, please contact JAO or EB for questions relative to the raw data.

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
