# Peer review of "Bartonella spp. Prevalence (Serology, Culture, and PCR) in Sanitary Workers in La Rioja Spain"

_pathogens, 2020, doi:10.3390/pathogens9030189_

Round 1

Reviewer 1 Report

Bartonella are globally distributed zoonotic bacterial pathogens causing a broad spectrum of clinical manifestations in humans and animals. Bartonella are highly prevalent in wild mammals throughout the world, in particular in rodents and bats. Infections in these reservoir hosts is typically asymptomatic. Bacteria reside in high numbers in the circulation and appear to be transmitted via ectoparasites or direct tissue trauma, also to the incidental hosts such as humans. Numerous recent reports describe isolation and characterization of new strains and species of Bartonella. Bartonella DNA, and in some cases viable bacteria, have also been detected in a number of different ectoparasites. In the clinic, Bartonella infections are rare, in particular in human, which contrasts the large environmental reservoir of Bartonella. However, Bartonella infections are actively debated as neglected or emerging infections. This is largely due to i) slow in vitro growth properties of Bartonella excluding the typical culture-based identification from patient samples, ii) relatively weak antibody responses complicating the typical serology-based identification from patient samples, iii) apparent chronic and stealthy course of infection with occasional peaks of bacteremia, and iv) general lack of awareness of Bartonella infections by clinicians and veterinarians.

Given this background, the manuscript of the authors is both timely and provides new information on the prevalence of Bartonella infections, in particular in a pool of apparently healthy study subjects.   

Below my specific comments.

The authors need to more thoroughly describe their study subjects. They speak about “sanitary workers”. What does this mean? In materials and methods, the authors describe that the study subjects were physicians, nurses, researchers, etc. working in the bioscience field. Have these study subjects had any previous exposure to experimental research with Bartonella, or do they work in the same buildings where the work / diagnostics on Bartonella is being conducted? This is an important point, given the findings and the arguments made by the authors at several points that their study subjects were expected to have limited exposure to Bartonella.   

The authors exaggerate the prevalence of bloodstream infection. Their argument is largely based on PCRs, although done after enrichment BAPGM culturing. The manuscript has to be thoroughly revised around this point. Agar plate isolate is the only solid way to show that the study subject is bacteremic, i.e. the bacterium in the blood has been viable. Other possible way would be to show the increase of Bartonella DNA copy-number with qPCR upon enrichment BAPGM culturing. This would show that Bartonella was viable at the point of blood sample taking and started to grow in BAPGM. I do not find a place in the manuscript where the authors would provide the qPCR copy number data. As for now, the data shows 4/100 bacteremic study subjects = 4 %.

The last chapter of the Discussion falls, in my opinion, a bit off from the main focus of the manuscript. Although the authors discuss interesting points regarding Bartonella transmission, the kind of text (for example, arguments on Bartonella being potentially facultative symbionts) would be more suited for a review article. The Discussion chapter is quite long, and would, in my opinion, benefit from shortening/focusing.  

I do not find GenBank accession numbers for all the MLST markers sequenced in this study.   

In Table 1 and Table 2, why do the authors add - (minus) before the %-values?

One spelling mistake spotted. Table 1 legend “comparaing”. Please, use spell check. 

Author Response

Enclosed please find a revised manuscript entitled “Bartonella spp. prevalence (serology, culture, and PCR) in sanitary workers in La Rioja Spain” by Aránzazu Portillo,  et. al, that is submitted for publication consideration in Pathogens.

We very much appreciate the overall positive assessment of our original manuscript submission. We also appreciate the comments, questions, and suggested edits from the reviewers. We have addressed each of the reviewer’s comments as follows:

Reviewer #1

Bartonella are globally distributed zoonotic bacterial pathogens causing a broad spectrum of clinical manifestations in humans and animals. Bartonella are highly prevalent in wild mammals throughout the world, in particular in rodents and bats. Infections in these reservoir hosts is typically asymptomatic. Bacteria reside in high numbers in the circulation and appear to be transmitted via ectoparasites or direct tissue trauma, also to the incidental hosts such as humans. Numerous recent reports describe isolation and characterization of new strains and species of BartonellaBartonella DNA, and in some cases viable bacteria, have also been detected in a number of different ectoparasites. In the clinic, Bartonella infections are rare, in particular in human, which contrasts the large environmental reservoir of Bartonella. However, Bartonellainfections are actively debated as neglected or emerging infections. This is largely due to i) slow in vitro growth properties of Bartonella excluding the typical culture-based identification from patient samples, ii) relatively weak antibody responses complicating the typical serology-based identification from patient samples, iii) apparent chronic and stealthy course of infection with occasional peaks of bacteremia, and iv) general lack of awareness of Bartonella infections by clinicians and veterinarians.

The authors very much agree with the above succinct and accurate assessment of our current understanding of the genus Bartonella and the disease bartonellosis.

Given this background, the manuscript of the authors is both timely and provides new information on the prevalence of Bartonella infections, in particular in a pool of apparently healthy study subjects.  

We appreciate this reviewer’s overall positive assessment of our manuscript submission.  

Below my specific comments.

The authors need to more thoroughly describe their study subjects. They speak about “sanitary workers”. What does this mean? In materials and methods, the authors describe that the study subjects were physicians, nurses, researchers, etc. working in the bioscience field. Have these study subjects had any previous exposure to experimental research with Bartonella, or do they work in the same buildings where the work / diagnostics on Bartonella is being conducted? This is an important point, given the findings and the arguments made by the authors at several points that their study subjects were expected to have limited exposure to Bartonella.   

This is a very good point. We have provided some additional detail regarding the study population in the subject recruitment section to better orient the reader. As described in the discussion, we did not expect to find such a high Bartonella seroprevalence or molecular prevalence. To date, occupational exposure risk has not been critically investigated for physicians, nurses and laboratory workers. Based upon our findings in this study, additional investigations are warranted.

The authors exaggerate the prevalence of bloodstream infection. Their argument is largely based on PCRs, although done after enrichment BAPGM culturing. The manuscript has to be thoroughly revised around this point. Agar plate isolate is the only solid way to show that the study subject is bacteremic, i.e. the bacterium in the blood has been viable. Other possible way would be to show the increase of Bartonella DNA copy-number with qPCR upon enrichment BAPGM culturing. This would show that Bartonella was viable at the point of blood sample taking and started to grow in BAPGM. I do not find a place in the manuscript where the authors would provide the qPCR copy number data. As for now, the data shows 4/100 bacteremic study subjects = 4 %.

We respectfully disagree with this conclusion. Although blood culture isolation is the historical “gold standard” for assessing bloodstream infection, the advent of various molecular-based diagnostic modalities continues to change our understanding of bloodstream infections. Although bacterial DNA can remain sequestered in sites such as dental pulp for centuries, bacterial and host DNA are constantly degraded and removed from the bloodstream. When we and others have experimentally infected animals with infectious organisms, PCR amplification of DNA occurs (with variable detection sensitivity) until the infection is immunologically or therapeutically eliminated, after which sequential PCR results are negative. For organisms that are even more difficult to culture than Bartonella spp. routinely in microbiology laboratories, such as Anaplasma, Babesia, Ehrlichia, and Rickettsia, or cannot be cultured (hemotropic Mycoplasma) PCR has become the predominant diagnostic modality used to assess infection status. Another way to look at this situation in the context of Bartonella spp. would be as follows: Would a physician act upon a positive PCR result confirmed by DNA sequencing in a patient with blood culture negative endocarditis prior to getting serology results? We would suggest that the answer is yes. As only 1 ml of blood is placed into 9 mls of BAPGM, those individuals who were only PCR+ after enrichment culture (PCR negative from blood) most likely had viable bacteria that grew in culture to a level of PCR detection. Due to the inherent difficulty of isolating Bartonella, we were surprised/shocked that we obtained 3 subculture isolates from seemingly healthy individuals. Due to the generally low level bacteremia, copy number is not particularly useful (if qPCR positive CT values are generally high).   

The last chapter of the Discussion falls, in my opinion, a bit off from the main focus of the manuscript. Although the authors discuss interesting points regarding Bartonella transmission, the kind of text (for example, arguments on Bartonella being potentially facultative symbionts) would be more suited for a review article. The Discussion chapter is quite long, and would, in my opinion, benefit from shortening/focusing.  

Agreed, we have deleted most of the last paragraph of the discussion and the cited references.

I do not find GenBank accession numbers for all the MLST markers sequenced in this study.  

In table 4, the DNA sequences from the three isolates obtained in this study were compared to  deposited GenBank accessions for 4 gene/spacer targets. Because all sequences had 100% similarity with published GenBank submissions and because we did not obtain any unique isolates or sequences as a result of this study, we did submit sequences to GenBank.   

In Table 1 and Table 2, why do the authors add - (minus) before the %-values?

Corrected.

spelling mistake spotted. Table 1 legend “comparaing”. Please, use spell check. 

Corrected

We hope that the editor and reviewers will find our revised manuscript to be acceptable for publication in Pathogens.   

Reviewer 2 Report

This is an interesting and well written paper which addresses a rather relevant topic to clinicians who take care of patients with Bartonella spp. infections. I have some minor revision to suggest:

Comments:

The discussion section is quite lengthy and I think it could be shortened without affecting the overall message.

Minor comments:

The presentation of Table 1 is somewhat confusing and could be improved.

  • First, the n participants are 97, but in the Table title it is 99. Please correct that.
  • Second, I suggest presenting the data in Table 1 in two separate columns, one for Bartonella PCR-positive and one for PCR-negative individuals and divide the numbers accordingly
  • Please describe U in a footnote, please write the CI in this fashion 95% CI and not CI95%
  • If there were no significant differences between the two groups with regard to the clinical features, show the numbers and write down under the P value column NS: non-significant for all.

Author Response

Enclosed please find a revised manuscript entitled “Bartonella spp. prevalence (serology, culture, and PCR) in sanitary workers in La Rioja Spain” by Aránzazu Portillo,  et. al, that is submitted for publication consideration in Pathogens.

We very much appreciate the overall positive assessment of our original manuscript submission. We also appreciate the comments, questions, and suggested edits from the reviewers. We have addressed each of the reviewer’s comments as follows:

Reviewer # 2

This is an interesting and well written paper which addresses a rather relevant topic to clinicians who take care of patients with Bartonella spp. infections. I have some minor revision to suggest:

We very much appreciate this reviewers overall positive assessment of our manuscript submission.

Comments:

The discussion section is quite lengthy and I think it could be shortened without affecting the overall message.

We have deleted most of the last paragraph of the discussion. The remaining discussion content seems pertinent for the reader’s assessment of the study results.

Minor comments:

The presentation of Table 1 is somewhat confusing and could be improved.

We have revised Table 1.

  • First, the n participants are 97, but in the Table title it is 99. Please correct that.

Corrected.

  • Second, I suggest presenting the data in Table 1 in two separate columns, one for Bartonella PCR-positive and one for PCR-negative individuals and divide the numbers accordingly

Modified as suggested.

  • Please describe U in a footnote, please write the CI in this fashion 95% CI and not CI95%

Modified as suggested

  • If there were no significant differences between the two groups with regard to the clinical features, show the numbers and write down under the P value column NS: non-significant for all.

One parameter, trekking was significant.

We hope that the editor and reviewers will find our revised manuscript to be acceptable for publication in Pathogens.